# CNS Superficial Siderosis Mimicking a Motor Neuron Disease

**DOI:** 10.3390/brainsci12111558

**Published:** 2022-11-16

**Authors:** Sergio Castro-Gomez, Julius Binder, Arndt-Hendrik Schievelkamp, Michael Thomas Heneka

**Affiliations:** 1Department of Neurodegenerative Disease and Geriatric Psychiatry/Neurology, University Hospital, 53127 Bonn, Germany; 2Institute of Physiology II, University Hospital Bonn, 53115 Bonn, Germany; 3Department of Neuroimmunology, Institute of Innate Immunity, University Hospital Bonn, 53127 Bonn, Germany; 4Department of Neuroradiology, University Hospital Bonn, 53127 Bonn, Germany; 5Luxembourg Centre for Systems Biomedicine, L-4365 Esch-sur-Alzette, Luxembourg

**Keywords:** case report, superficial siderosis, hemosiderin accumulation, motoneuron disease, spinal MRI, ALS-mimics

## Abstract

Superficial siderosis of the central nervous system (SS-CNS) is a rare condition characterized by a hemosiderin accumulation along the subpial surfaces and arises from an intermittent chronic bleeding in the subarachnoid space usually as a result of a chronic subarachnoid hemorrhage by trauma, vascular malformations, CNS tumors, or cerebral amyloid angiopathy (CAA). We present a 61-year-old male with a 12-year history of limb weakness, muscle wasting, cramps, clumsiness, progressive unsteady gait, and fine motor impairments. His medical history included the resection of a left parietal meningioma and a myxopapillary ependymoma near the conus terminalis (L3/4) at the age of 51 years. The clinical examination revealed a motor neuron syndrome with a clear bilateral wasting of the hand muscles, a diffuse atrophy of the shoulder and calf muscles, and a weakness of the arms, fingers, hips, and feet. Deep tendon reflexes were symmetrically briskly hyperactive. Standing and walking were only possible with a support. Magnetic resonance imaging of the entire neuroaxis showed progressive severe cerebral, brainstem, and spinal superficial siderosis in form of extensive hypointensities on T2-weighted gradient-echo images and susceptibility-weighted sequences. Despite a successful neurosurgical removal of the tumors and delaed medical treatment with an iron chelator for one year, we observed no clinical recovery or stability in our patient, making this case unique, and suggesting an irreversible neurodegenerative process. This case reinforces the need of including SS-CNS in the list of amyotrophic lateral sclerosis (ALS)-mimics and demonstrates the fundamental use of a complete neuraxial MRI investigation on evaluating possible ALS cases.

## 1. Introduction

Superficial siderosis of the central nervous system (SS-CNS) is a rare condition characterized by a hemosiderin accumulation along the subpial surfaces that induces a clinical syndrome typically characterized by a progressive cerebellar ataxia, sensorineuronal deafness, and pyramidal signs [1]. SS-CNS occurs by intermittent chronic bleeding in the subarachnoid space usually as the result of a chronic subarachnoid hemorrhage by trauma, vascular malformations, CNS tumors, or cerebral amyloid angiopathy (CAA) [2]. Magnetic resonance imaging (MRI) is the preferred method to identify SS-CNS and possible sources of hemorrhage. Iron deposits cause paramagnetic effects on T2-weighted and gradient-echo images showing typical “black rims”. The evolution of the SS-CNS is poorly understood and the removal of an established source of bleeding does not always bring a clinical recovery or the reversal of the SS. Additionally, in a third of the cases, the source of the bleeding remains undetermined. In this report, we describe a patient with a severe SS-CNS and an atypical slow progressive motor neuron syndrome, who was admitted to our department with suspected amyotrophic lateral sclerosis (ALS). His history included a meningioma located in the left parietal region and a myxopapillary ependymoma near the conus terminalis (L3/4) (Figure 1a,b). Despite the successful neurosurgical removal of the tumors, an exhaustive angiographic examination, and delayed medical treatment with an iron chelator for one year, no source of the bleeding, a clinical recovery, or stability were observed. This case reinforces the need and relevance for a complete neuraxial MRI investigation in patients with suspected ALS or with a motoneuron syndrome that not completely fulfill the ALS criteria in order to identify disorders mimicking this disease.

## 2. Case Presentation

We evaluated a 61-year-old male for a 12-year history of progressive unsteady gait. This was accompanied by a progressive limb weakness and muscle wasting (proximal more than distal, lower more than upper limbs), cramping in the legs, general clumsiness, and fine motor impairments. The patient described being able to walk only with a walking frame and being incapable of mastering stairs. His medical history included the removal of a meningioma (WHO Grade I) located in the left parietal region and a successful resection of a myxopapillary ependymoma (WHO Grade I) near the conus terminalis (L3/4) at the age of 51 years (Figure 1). The perioperative neurological evaluation described a spastic ataxic gait without any other focal deficits or abnormal reflexes (estimated ALSFRS-R of 46/48). A further clinical examination 2 years after surgery showed no clinical changes in the neurologic status and he experienced progressive motor and gait dysfunction over the subsequent years without a rapid deterioration at any time. He also had a history of smoking (approx. 30 packs a year), an untreated hypertension, and an urge incontinence associated with symptomatic benign prostatic hyperplasia (BPH). His medications included Tamsulosin. His father was diagnosed with late-onset Parkinson’s disease.

**Figure 1 brainsci-12-01558-f001:**
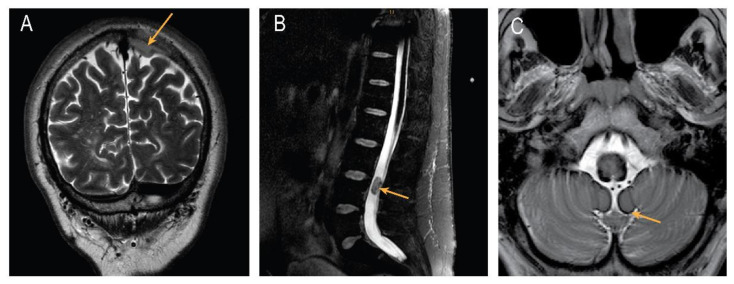
Brain and spinal magnetic resonance imaging (MRI). (**A**) Coronal T2-weighted imaging showing left parietal meningioma; (**B**) sagittal T2-weighted imaging using Spectral Attenuated Inversion Recovery (SPAIR) of a myxopapillary ependymoma near the conus terminalis (L3/4); (**C**) T2-weighted images of cerebellum and brainstem showing superficial siderosis.

At the time of presentation to our department, the clinical examination revealed a clear bilateral wasting of the abductor pollicis brevis, first dorsal interosseous muscle (thenar muscles), and abductor digiti minimi (ADM) (hypothenar muscle). There was a bilateral diffuse atrophy of the shoulder and calf muscles. Manual muscle strength tests demonstrated a weakness according to the Medical Research Council scale (right/left) in the following muscle groups: arm flexion (5/4 M. biceps brachii), hand-interossei (4/4) at the spreading of the fingers, hip flexors (4/4), and the flexion–extension of the feet (3/3). Deep tendon reflexes were symmetrically brisk in the upper and lower limbs with the spread of contraction across the adjacent joints and normal plantar responses. A sensory examination showed a reduced sense of vibration at both medial malleoli (four points on a semiquantitative eight-point scale measured with a Rydel–Seiffer tuning fork). Standing and walking were only possible with a support. He had a wide-based gait with difficulty walking tandem. The patient and his wife reported no hearing loss, bulbar or respiratory symptoms, behavioral, or cognitive changes. A neuropsychological examination including a CERAD test battery showed a mild cognitive disorder of multiple domains (memory, language, attention, and visual–spatial function), and the Edinburgh Cognitive ALS Screen (ECAS) shows abnormal values for non-ALS-specific parameters (executive functions, memory, and visual–spatial function). However, the ALS-specific parameters (speech and fluency) were unremarkable. The patient had 31 points on the Revised ALS Functional Rating Scale (ALSFS-R). The laboratory studies were normal except for an elevated Creatine kinase level of 253 U/L (normal level <190 U/L) and a Creatine kinase level–Myocardial Band level of 5.7 ng/mL (normal level <4.87 U/L). Electrodiagnostic testing showed a normal sensory nerve conduction of the right median, right ulnar, and right sural nerve. Motor nerve conduction studies suggested a reduced compound muscle action potential (CMAP) amplitude in the right peroneal nerve and left tibial nerve without evidence of a conduction block. The F-wave latencies of the right median nerve, ulnar nerve on both sides, peroneal nerve on both sides, and left tibial nerve were normal. The needle electromyography of the left extensor digitorum communis, right tibialis anterior, and left vastus lateralis muscles demonstrated few fibrillation potentials and a prolonged motor unit potential duration. The thoracic paraspinal and deltoideus muscles presented no spontaneous activity. The transcranial magnetic stimulation showed a prolonged central motor conduction time in both the ADM and both tibialis anterior muscles.

MRIs of the entire neuroaxis were performed twice in a 10-year span, showing slowly progressive cerebral and spinal superficial siderosis (Figure 1c and Figure 2a–d).

A lumbar puncture 2 years after the disease’s onset was reported as normal (without sings of bleeding or positive ferritin). A hypointense venous line in the anterior spinal canal area was initially detected. In a digital subtraction angiography (DSA), this line was described as a prevertebral vein, most likely fed by a branch of the left-sided posterior auricular artery. In 2019, we performed new MRIs of the entire neuroaxis to evaluate the progression and detect any potential additional sources of hemorrhage. The brain MRI showed the parietal dural defect after meningioma extirpation and a global atrophy. Moreover, it showed a progressive basal, intraventricular, and superficial siderosis in the entire central nervous system in the form of extensive hypointensities on T2-weighted gradient-echo images, fluid-attenuated inversion recovery (FLAIR), and susceptibility-weighted imaging (SWI) sequences (Figure 2a–d). There was no evidence of aneurysms, vessel malformations, or active sources of bleeding. The hemosiderosis in the brain and spinal cord was described as massive and far more pronounced than in 2010. No intraspinal fluid-filled collections were found. We performed an additional DSA without evidence of further vascular pathology or the source of the bleeding. The patient was discharged from our department with the chelating agent deferiprone at a dose of 750 mg t.i.d. in an attempt to reduce the symptoms caused by the iron deposition. One-year-long therapy with deferiprone showed no significant effects in the clinical evolution.

## 3. Discussion

The SS-CNS results from an accumulation of hemosiderin on the subpial layer of the brain and spinal cord. Clinically, it typically presents with progressive gait ataxia and a neurosensorial hearing impairment. Other, less frequent clinical features include dementia, neurogenic bladder dysfunction, anosmia, anisocoria, sensory signs, extraocular motor palsies, backache, sciatica, and lower motor neuron signs. SS-CNS is caused by the chronic or intermittent extravasation of blood into the subarachnoid space, and the dissemination of the heme by circulating cerebrospinal fluid, usually in the course of a chronic subarachnoid hemorrhage, including trauma, vascular malformations, CNS tumors, or CAA [2,3]. Pathologically extensive deposits of iron and ferritin, perivascular collections of hemosiderin-laden macrophages, and iron-positive anuclear foamy structures in the neuropil that resemble axonal spheroids are found along the subpial surfaces [4]. Only in a third of the cases can the source of the hemorrhage be identified. Since the implementation of MRI in the clinical practice, the identification of the SS-CNS has increased, yet the overall prevalence remains low. The characteristic MRI finding is a rim of marked hypointensity on T2-weighted or SWI images surrounding the brain stem, spinal cord, sylvian and interhemispheric fissures, and a few cortical sulci. Frequently, intraspinal fluid-filled collections (meningoceles, pseudomeningoceles, epidural cysts, arachnoid cysts, or cerebrospinal fluid [CSF] loculations) are also found and thought to be related to disease pathogenesis, since the repair of the dural defects seems to be accompanied by a clinical stability or an improvement in some cases [5]. In this report, we present a patient with an unusually slow progressive motor neuron syndrome (proximal muscle atrophy, tetraparesis, and brisk reflexes) together with a progressive severe SS-CNS. In the medical history, a meningioma located in the left parietal region and a myxopapillary ependymoma near the conus terminalis were removed two years after the onset of symptoms. In this patient, minor sensory changes were detected on the clinical examination which likely originated from the treated ependymoma. Thereafter, we carefully followed the patient over 10 years. Despite the surgical treatment of the initially suspected bleeding sources, the SS-CNS and the motor neuron syndrome kept progressing slowly without a bulbar or respiratory impairment, and an exhaustive angiographic examination could not find any source of the bleeding. It should be noted that SS-CNS has not been considered a differential diagnostic of ALS [6] and to our knowledge, only few other cases with a similar clinical phenotype have been reported in the literature until now [7,8,9,10,11]. Interestingly, and in contrast to our patient, the reports by Driver-Dunckley et al. [8], Payer et al. [9], Kumar et al. [10], and Deguchi et al. [11] found intraspinal fluid-filled collections and chronic CSF leaks to be the possible bleeding source. In these cases, a neurosurgical correction of the epidural collections brought a transient or no improvement. Moreover, the authors reported only short follow-ups questioning the real effectiveness of the aforementioned interventions in a long run. Our patient underwent surgical treatment 2 years after the onset of their symptoms; this intervention showed no effect on the progression of the disease. The first long-term prospective study with 38 cases treated with the iron chelator deferiprone showed a measurable reduction in the hemosiderin and stabilization or improving the course of the disease in about half of the patients [12]. Accordingly, our patient was treated with deferiprone for over 1 year with neither evidence for clinical stability nor a supratentorial hemosiderin reduction in the MRIs. Interestingly, during the 10-year follow-up, our patient showed no typical signs of sensorineural deafness and diagnostic studies failed to find persistent sources of bleeding at the latest time point, making this case unique and suggesting an irreversible neurodegenerative process.

The pathophysiological mechanism of neurodegeneration triggered by SS-CNS remains poorly understood. Studies on the post-mortem tissue have shown that the parenchymal uptake of CSF circulating heme and its subsequent conversion into iron or hemosiderin is partly mediated by radial glial cells (e.g., Bergmann glia in the cerebellum). Hemosiderin accumulates chronically in granules that may release iron from their protein matrix. In contrast to other organs, the CNS is unable to fully clear iron, which itself could oxidize lipids and other neuronal structures. Hemolytic products and CSF heme also trigger the expression of heme-oxygenase-1 (HO-1) and ferritin in radial glia cells, microglia, and infiltrating macrophages [13]. The role of microglia in other neurogenerative diseases has been extensively discussed [14], but little is known about its role in SS-CNS. The treatment with the HO inhibitor tin-protoporphyrin in experimental SS-CNS in rabbits has shown to prevent a hemosiderin deposition, microglia proliferation, and ferritin translation in these cells [15]. Future, experimental, translational, and clinical studies are needed to better understand the natural history, pathophysiology, and treatment of the SS-CNS.

## 4. Conclusions

We here presented a long follow-up of a rare case of a severe SS-CNS manifested with a motor neuron syndrome. This case report is unique and reinforces the need for a complete neuraxial MRI investigation of possible ALS cases to identify SS-CNS in early stages. Additional to the early treatment of possible hemorrhage sources, a combined treatment with an iron chelator and a neuroinflammatory modulator might be necessary to prevent a secondary neurodegeneration due to SS-CNS.

## Figures and Tables

**Figure 2 brainsci-12-01558-f002:**
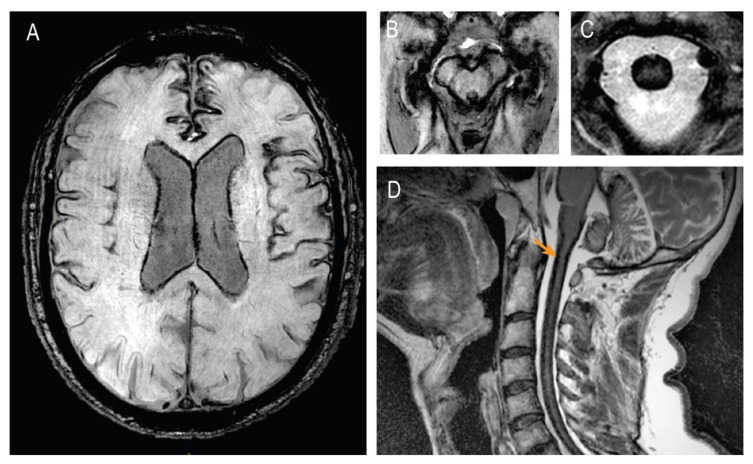
Follow-up brain and spinal MRI showing severe SS-CNS. (**A**) Susceptibility weighted transverse image SWI showing severe cortical and periventricular SS; (**B**) SWI of mesencephalic structures; (**C**) SWI of cervical spinal cord; (**D**) T2-weighted gradient-echo sagittal images of cervical spinal cord showing severe spinal SS.

## Data Availability

Not applicable.

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
