# Peer review of "CNS Superficial Siderosis Mimicking a Motor Neuron Disease"

_brainsci, 2022, doi:10.3390/brainsci12111558_

Round 1

Reviewer 1 Report

This is an interesting case report on a single patient presenting with pronounced superficial siderosis and signs of a motoneuron disease.

A few points, however, need additional attention:

- The clinical presentation and progression is not clear. Apparently, the authors describe the patient at the age of 61, but say that he had progressive unsteadiness of gait for 12 years. Meningeoma and ependymoma were removed at age 51, thus already two years after symptom onset? Is there any neurological status recorded at that time? Can authors reconstruct the ALSFRS-R for the initial examination? Was there any clinical examination in between? Did the patient always show a continuous progression or any rapid deterioration?

- Authors mention that there are two MRIs showing progression of the siderosis over a period of 10 years, but figure 2 only shows the follow-up MRI. Please add images of the first MRI.

- The EMG report is unclear: "fewer fibrillations with a high and prolonged amplitude" are mentioned. Do authors mean MUAP? Fibrillations would be a sign of acute degeneration.

- Nerve conduction studies should be mentioned in detail: was a reduction in amplitude or a prolongation of the nerve conductance velocity observed, or both? Any conduction blocks?

- Did authors test for the presence of GM1 antibodies in this patient?

- Authors mention a lumbar puncture. Did they determine the value of neurofilament light chain or phosphorylated heavy chain? If not in CSF, these values could also be obtained from serum samples.

- Neuropsychological assessment is mentioned, but not further quantified. Please specify.

- Authors write that the clinical presentation is "caused by a profound SS-CNS". This is an assumption, which cannot be proven by the authors. How can authors be sure that this is indeed the cause of the symptoms? On the same line, why do authors conclude that the tumours are causative. As they themselves write, after tumour removal, the disease progressed and the siderosis did not improve. Meningeomas are very frequently observed, but rarely accompanied by siderosis. Could it be a mere coincidence? Throughout the manuscript, please replace causality by coincidence.

- How do authors explain that the patient has no bulbar symptoms at all, although the siderosis clearly affects the midbrain and pons?

- What was the rationale of treating the patient with Deferiprone for one year? According to the authors, the progression was slow. How fast would authors expect a clinical effect?

- If the patient formally passed the diagnostic criteria of ALS, why was he not treated with Riluzole?

- Did authors perform any genetic testing on this patient?

- Please specify the exact designation of the MRI sequences in the legends, e.g., missing in Fig. 1

- Line 114: Fig. 2 is meant?

- The reviewer is puzzled about the use of the term "study" in the authors contributions as well as in regard to the IRB statement: this appears to be a case report of a patient, who was treated on a regular basis. Why do authors call this a "study"? Why did the authors involve the Ethics commission? Deferiprone treatment was used off-label, but this would not require to call upon EC. Please explain.

Author Response

Dear Mr. Eric Yu

Thank you for considering our manuscript and thank you to the reviewers for the insightful and constructive comments. We have revised our manuscript to address the points raised by the two reviewers. Please find below our point-by-point response.

As requested, the revised manuscript is in the track-changes mode.

Response to Reviewer 1 Comments

This is an interesting case report on a single patient presenting with pronounced superficial siderosis and signs of a motoneuron disease.

A few points, however, need additional attention:

- The clinical presentation and progression is not clear. Apparently, the authors describe the patient at the age of 61, but say that he had progressive unsteadiness of gait for 12 years. Meningeoma and ependymoma were removed at age 51, thus already two years after symptom onset? Is there any neurological status recorded at that time? Can authors reconstruct the ALSFRS-R for the initial examination? Was there any clinical examination in between? Did the patient always show a continuous progression or any rapid deterioration?

Response: The reviewer raises a very important point. The perioperative neurological evaluation described a spastic ataxic gait without any other focal deficits or abnormal reflexes not explained by the tumors’ topology (retrospective ALSFRS-R of 46/48). A further clinical examination 2 years after surgery showed no clinical changes in the neurologic status and he experienced progressive motor and gait dysfunction over the subsequent years. The patient showed no rapid deterioration at any time, the progressive symmetrical paraparesis and motor impairment were described as continuous. This information was also added to the main text.

- Authors mention that there are two MRIs showing progression of the siderosis over a period of 10 years, but figure 2 only shows the follow-up MRI. Please add images of the first MRI.

Response: Prompted by the reviewer's suggestion, a T2 weighted image was added to figure 1c. showing the superficial siderosis initially described

- The EMG report is unclear: "fewer fibrillations with a high and prolonged amplitude" are mentioned. Do authors mean MUAP? Fibrillations would be a sign of acute degeneration.

Response: Needle electromyography of the extensor digitorum communis, tibialis anterior and vastus lateralis muscles demonstrated few fibrillation potentials and prolonged motor unit potential duration. The sentence was also amended in the main text.

- Nerve conduction studies should be mentioned in detail: was a reduction in amplitude or a prolongation of the nerve conductance velocity observed, or both? Any conduction blocks?

Response: Motor nerve conduction studies suggested decreased compound muscle action potential (CMAP) amplitude in the right peroneal nerve and left tibial nerve without evidence of conduction block. The sentence was also changed in the main text.

- Did authors test for the presence of GM1 antibodies in this patient?

Response: Anti-GM1 antibodies were not tested since the clinical and electrodiagnostic criteria for a multifocal motor neuropathy were not fulfilled, especially if diffuse symmetrical weakness during the initial weeks is taken as exclusion criteria.

- Authors mention a lumbar puncture. Did they determine the value of neurofilament light chain or phosphorylated heavy chain? If not in CSF, these values could also be obtained from serum samples.

Response: Although this measurement could be very interesting in this case, the neurofilament light chain or phosphorylated heavy chain in CSF were not determined since the lumbar puncture were performed almost 13 years ago, when these biomarkers were still under incipient research in the motoneuron field, diagnostic value was not universally accepted and the test availability was limited.

Certainly, the neurofilament light chain can be measured in serum. This measurement would take us at least a month longer as the patient should be contacted and seen in our department for blood examination. Due to the progressive neurological deterioration, we would expect an elevation of sNFL as it is seen in other neurological disorders associated with neuronal death (e.g. HIV) and therefore it wouldn't give us any specific diagnostic information.

- Neuropsychological assessment is mentioned, but not further quantified. Please specify.

Response: The CERAD test battery showed a mild cognitive disorder of multiple domains (memory, language, attention, visual-spatial reasoning). The Edinburgh Cognitive ALS Screen shows abnormal values for non-ALS-specific parameters (executive functions, memory, and spatial imagination). However, ALS-specific parameters (speech, fluency) were unremarkable. An ALS-specific pattern of cognitive deficits could not be confirmed. Prompted by the reviewer's suggestion, the sentence was also changed in the main text.

- Authors write that the clinical presentation is "caused by a profound SS-CNS". This is an assumption, which cannot be proven by the authors. How can authors be sure that this is indeed the cause of the symptoms? On the same line, why do authors conclude that the tumours are causative. As they themselves write, after tumour removal, the disease progressed and the siderosis did not improve. Meningeomas are very frequently observed, but rarely accompanied by siderosis. Could it be a mere coincidence? Throughout the manuscript, please replace causality by coincidence.

Response: We agree with the reviewer and made the respective changes in the main text. 

- How do authors explain that the patient has no bulbar symptoms at all, although the siderosis clearly affects the midbrain and pons?

Response: This is an excellent question, for which we unfortunately do not have a certain answerer. We speculate that in this case the corticobulbar tract and brainstem nuclei such as the nucleus ambiguous or nucleus hyoglossus involved in the bulbar functions are not severely affected since less hemosiderin signal was found in the fourth ventricle and these structures are found deeper than the lateral corticospinal tract and motoneurons located in the spinal cord that are in close contact with severe hemosiderin deposits being more susceptible to degeneration.

- What was the rationale of treating the patient with Deferiprone for one year? According to the authors, the progression was slow. How fast would authors expect a clinical effect?

Response: We based our treatment on a pilot safety trial by Levy and Llinas 2012, where the treatment with deferiprone for three months proved to be safe and in some of the patients showed evidence of reduced hemosiderin deposition, clinically improvement or stability (1).

  1. Levy M, Llinas R. Pilot Safety Trial of Deferiprone in 10 Subjects With Superficial Siderosis. Stroke. 2012 Jan;43(1):120–4.

- If the patient formally passed the diagnostic criteria of ALS, why was he not treated with Riluzole?

Response: in 2018, we recommended treatment with Riluzole and Rasagilin. These treatments were not taken by the patient due to the uncertainty of the ALS diagnosis.

- Did author perform any genetic testing on this patient?

Response: No genetic testing was performed.

- Please specify the exact designation of the MRI sequences in the legends, e.g., missing in Fig. 1 te

Prompted by the reviewer's suggestion, the following information was added to the figure legend: Figure 1. (a) Coronal T2-weighted imaging showing left parietal meningioma; (b) Sagital T2-weighted imaging using SPectral Attenuated Inversion Recovery (SPAIR) of a myxopapillary ependymoma near the conus terminalis (L3/4); (c). T2-weighted images of cerebellum and brainstem showing superficial siderosis.

- Line 114: Fig. 2 is meant?

Response: yes, Fig 2 is meant. A correction to the text was accordingly made.

- The reviewer is puzzled about the use of the term "study" in the authors contributions as well as in regard to the IRB statement: this appears to be a case report of a patient, who was treated on a regular basis. Why do authors call this a "study"? Why did the authors involve the Ethics commission? Deferiprone treatment was used off-label, but this would not require to call upon EC. Please explain.

Response: We agree with the reviewer and made the respective changes to the word “study”. The Ethics commission was formally involved to evaluate whether the publication of this case carried any ethical issues. The Ethics Committee confirmed that the publication of this case report did not raise any ethical or professional concerns, as it was an analysis of retrospective data in the context of routine diagnosis and treatment. 

Reviewer 2 Report

In this report the authors aimed to present a case of a patient with slowly progressive ALS phenotype (probable disease as categorized by the El Escorial criteria) in whom MRI identified diffuse CNS superficial siderosis.

This case is interesting but a few points should be improved.

In introduction, it should be clear that neuroaxial MRI investigation is relevant in patients without a definite clinical diagnosis of ALS.

In introduction, the source of bleeding is not found in one third of the cases, but they write the opposite in Discussion.

In this patient minor sensory changes were detected on clinical examination, but these changes could be originated from the treated ependymoma not being related to the superficial siderosis, important to stress this in discussion.

There is no information regarding respiratory function.

Mild cognitive impairment is too vague, do they mean a mild mnesic dysfunction or an executive dysfunction.

They write “motor conduction suggested damage of right common peroneal nerve and left tibial nerve”, this is very unclear, were the sensory potentials of these nerves abnormal suggesting a nerve trunk lesion, or simply the motor amplitudes were decreased due to the loss of motor units? Regarding needle EMG changes, were the mentioned muscles affected bilaterally or unilaterally? Were thoracic and bulbar muscles investigated?

Presenting TMS data absent responses were observed recording from ADM, it should be shared the CMAP amplitudes of ADM and tibialis anterior muscles.

In discussion, how to explain the urinary symptoms? In discussion the authors suggested that this patient had a motor neuron disease cause by the toxicity associated with superficial siderosis, which is different from discussing a mimicking disorder. The authors need to be clear.

Author Response

Dear Mr. Eric Yu

Thank you for considering our manuscript, and thank you to the reviewers for the insightful and constructive comments. We have revised our manuscript to address the points raised by the two reviewers. Please find below our point-by-point response.

As requested, the revised manuscript is in the track-changes mode.

Response to Reviewer 2 Comments

Comments and Suggestions for Authors

In this report, the authors aimed to present a case of a patient with slowly progressive ALS phenotype (probable disease as categorized by the El Escorial criteria) in whom MRI identified diffuse CNS superficial siderosis. This case is interesting, but a few points should be improved.

- In introduction, it should be clear that neuroaxial MRI investigation is relevant in patients without a definite clinical diagnosis of ALS.

Response: we thank the reviewer for the positive and helpful comments. Prompted by the reviewer's suggestion, we added the following information to the introduction: This case reinforces the need and relevance for a complete neuraxial MRI investigation in patients with suspected ALS or with a motoneuron syndrome that not completely fulfills the ALS criteria in order to identify disorders mimicking this disease.

- In introduction, the source of bleeding is not found in one third of the cases, but they write the opposite in Discussion.

Response: We agree with the reviewer, in our case a clear source of bleeding was never found, and the tumors cannot be considered as one of them.

We amended this sentence to the introduction: Despite successful neurosurgical removal of the tumors, exhaustive angiographic examination and delayed medical treatment with an iron chelator for one year, no source of bleeding, clinical recovery or stability was observed. We added this sentence to the introduction.

- In this patient minor sensory changes were detected on clinical examination, but these changes could be originated from the treated ependymoma not being related to the superficial siderosis, important to stress this in discussion.

Response: We thank the reviewer for the helpful comments. This point was additionally added to the discussion.

-There is no information regarding respiratory function.

Response: During the whole follow-up (10 year), the patient presented no respiratory complaints. This information was also added to the clinical description.

- Mild cognitive impairment is too vague, do they mean a mild mnesic dysfunction or an executive dysfunction.

Response: The CERAD test battery showed a mild cognitive disorder of multiple domains (memory, language, attention, visual-spatial reasoning). The Edinburgh Cognitive ALS Screen shows abnormal values for non-ALS-specific parameters (executive functions, memory, and spatial imagination). However, ALS-specific parameters (speech, fluency) were unremarkable. An ALS-specific pattern of cognitive deficits could not be confirmed. Prompted by the reviewers´ suggestion, the sentence was also changed in the main text.

- They write “motor conduction suggested damage to right common peroneal nerve and left tibial nerve”, this is very unclear, were the sensory potentials of these nerves abnormal suggesting a nerve trunk lesion, or simply the motor amplitudes were decreased due to the loss of motor units?

- Regarding needle EMG changes, were the mentioned muscles affected bilaterally or unilaterally? Were thoracic and bulbar muscles investigated?

Response: Regarding electrodiagnostic testing, we complemented and amended the whole section.

“Electrodiagnostic testing showed normal sensory nerve conduction of the right median, right ulnar and right sural nerve. Motor nerve conduction studies suggested reduced compound muscle action potential (CMAP) amplitude in the right peroneal nerve and left tibial nerve, without evidence of conduction block. F-wave latencies of the right median nerve, ulnar nerve on both sides, peroneal nerve on both sides and left tibial nerve were normal. Needle electromyography of the left extensor digitorum communis, right tibialis anterior and left vastus lateralis muscles demonstrated few fibrillation potentials and prolonged motor unit potential duration. Thoracic paraspinal and deltoideus muscles presented no spontaneous activity. Transcranial magnetic stimulation showed prolonged central motor conduction time in the left ADM muscle and in both tibialis anterior muscles.

- Presenting TMS data absent responses were observed recording from ADM, it should be shared the CMAP amplitudes of ADM and tibialis anterior muscles.

Response: we thank the reviewer for the helpful comment. We review the clinical record and complemented the sentence as follows:

Transcranial magnetic stimulation showed prolonged central motor conduction time in both ADM and both tibialis anterior muscles.

TMS (CMCT in ms) Muscle                 ADM re     ADM li                  TA re         TA li

Total motor time:                                 27,1         23,7                     43,5            42,9                  

                                                                  (max. 23 )                           (max. 31,9 )

Central motor time:                              11,5           8,5                      20,4             21,7                  

                                                               (max. 8,3 )                             (max. 16,3 )

- In discussion, how to explain the urinary symptoms?

Response: We thank the reviewer for this constructive comment. After checking again, the medical records, it was described an urge incontinence associated with symptomatic benign prostatic hyperplasia (BPH), probably unrelated to the neurological condition.

- In discussion, the authors suggested that this patient had a motor neuron disease cause by the toxicity associated with superficial siderosis, which is different from discussing a mimicking disorder. The authors need to be clear.

Response: We propose that the motoneuron syndrome presented by this patient could be initially misinterpreted as ALS, given the observed clinical phenotype.

Round 2

Reviewer 1 Report

The authors have now significantly improved their manuscript and answered all queeries.

Reviewer 2 Report

This reviewer is pleased with the current version